# Hippo Signaling in the Endometrium

**DOI:** 10.3390/ijms23073852

**Published:** 2022-03-31

**Authors:** Sohyeon Moon, Semi Hwang, Byeongseok Kim, Siyoung Lee, Hyoukjung Kim, Giwan Lee, Kwonho Hong, Hyuk Song, Youngsok Choi

**Affiliations:** Department of Stem Cell and Regenerative Biotechnology, Konkuk University, Seoul 05029, Korea; 1004sh.moon@gmail.com (S.M.); tpal34567@naver.com (S.H.); qufsksrlaqud@naver.com (B.K.); yis1717@konkuk.ac.kr (S.L.); mykhj102@naver.com (H.K.); gyaan@naver.com (G.L.); hongk@konkuk.ac.kr (K.H.); songh@konkuk.ac.kr (H.S.)

**Keywords:** Hippo signaling, uterus, estrogen, progesterone, estrous cycle

## Abstract

The uterus is essential for embryo implantation and fetal development. During the estrous cycle, the uterine endometrium undergoes dramatic remodeling to prepare for pregnancy. Angiogenesis is an essential biological process in endometrial remodeling. Steroid hormones regulate the series of events that occur during such remodeling. Researchers have investigated the potential factors, including angiofactors, involved in endometrial remodeling. The Hippo signaling pathway discovered in the 21st century, plays important roles in various cellular functions, including cell proliferation and cell death. However, its role in the endometrium remains unclear. In this review, we describe the female reproductive system and its association with the Hippo signaling pathway, as well as novel Hippo pathway genes and potential target genes.

## 1. Introduction

Cellular differentiation, proliferation, and tissue size control have long been mysteries in developmental biology. Molecular mechanisms regulate tissue growth during development and regeneration. In such a growth control system, tissue size must be communicated to each cell to regulate cell proliferation. Several genes that control both cell proliferation and apoptosis have been identified. The discovery of the Hippo signaling pathway helps us to understand and solve these riddles and mechanisms. The Hippo pathway was first discovered [1,2,3,4,5]. It is an evolutionarily conserved signaling cascade that regulates numerous biological processes, including cell growth, fate, and determination, tissue size control and regeneration, as a major regulator of tissue growth [6,7]. Various biochemical, physical, and structural signals and, stressors, including mechanical signals, cell polarity, cell-cell adhesion, hormones, and biologically active chemicals influence this pathway. Such a part of the Hippo signaling cascade is related to the plasma membrane, and substances in the extracellular environment or cell-to-cell interaction can trigger a complex network of regulatory mechanisms [8,9,10].

The uterus is an important part of the reproductive system and is precisely regulated by female hormones for the initiation and maintenance of pregnancy [11]. Estrogen plays a role in regulation of proliferation of the uterine endometrium. The estrogen receptor (ER) acts as a transcription factor that directs target gene expression, and it exerts this via a membrane bound ER. The latter is involved in regulating gene expression using intracellular signaling systems such as the mitogen-activated protein kinase (MAPK/ERK), phosphoinositide 3-kinase (PI3K) and RAS signaling pathways [11]. Many signaling pathways are linked to each other in endometrial dynamics.

Recently, several studies have reported on the Hippo signaling pathway in the uterus. This review summarizes the implications of the Hippo pathway as a new signaling system in the physiology and pathology of the uterus.

## 2. Hippo Signaling in Various Cellular Functions

### 2.1. Kinase Reaction Related to Hippo Signaling Pathway in Cellular Functions

In the standard Hippo kinase cascade, the STK3/4-SAV1 complex phosphorylates and activates the LATS1/2-MOB1A/B complex, which in turn phosphorylates and inactivates YAP/TAZ [12]. Recent work has not only described detailed molecular mechanisms in the standard Hippo kinase cascade but has also identified kinases and phosphatases that participate in the cascade (Figure 1). Activation of STK3/4 requires the phosphorylation of threonine 180/183 within the activation loop. STK3/4 can be activated by TAOK1-3 [13,14,15]. STK3/4 phosphorylates LATS1/2, which is promoted by SAV1, MOB1A/B, and NF2. Phosphorylated LATS1/2 phosphorylates YAP at serine 127 (YAP[S127]) and induces its sequestration in the cytoplasm. Cytoplasmic isolation of YAP through interaction with 14-3-3 proteins inhibits cell proliferation and tissue growth by inhibiting anti-apoptotic programs [16]. Mutation of a key component of the Hippo pathway or overexpression of YAP increases the level of functional YAP in the nucleus, thereby maintaining proliferation and anti-apoptotic transcription programs, and promoting cancer development by overcoming normal cell-cell contact inhibition and tissue size control [17,18]. The Hippo signaling component is essential for regulating YAP phosphorylation and activity in mammalian cells. However, STK3/4 knockout mouse embryonic fibroblasts (MEF) exhibit normal LATS1/2 phosphorylation. Cell-cell contact induces YAP(S127) phosphorylation and its maintenance in the cytoplasm and is important for contact inhibition of cell proliferation. The phosphorylation of cell contact-induced YAP(S127) in STK3/4 null MEF is not impaired [19,20]. Like YAP(S127) phosphorylation, LATS1/2 activation loop phosphorylation is induced at high cell density regardless of the state of STK3/4 [21]. As an alternative to LATS1/2 activation by STK3/4, P190A phosphorylates LATS1/2. The major GTPase activating protein (GAP) of the Rho family, P190A RhoGAP, is encoded by ARHGAP35 [22]. Rho GTPases have roles in signaling pathways that regulate various cellular activities, including gene expression, cell cycle progression, counting and skeletal reorganization and are essential regulators of tissue morphogenesis during embryonic development (Figure 2). They also play an important role in cancer development and progression, including unlimited proliferation, apoptosis avoidance and survival, and tissue metastasis and invasion [23]. p190 RhoGAP knockout mice were found to be an important regulator of cell size in developing mice as body size decreases and cell size also decreases [24]. In another study, P190A promoted the contact inhibition of proliferation (CIP) in cells, a fundamental characteristic of cancer cells in Madin-Darby Canine Kidney (MDCK) epithelial cells [25]. YAP is a well-established regulator of CIP. P190A induces CIP by inhibiting the expression of a gene cassette controlled by YAP through the activation of LATS1/2 in MDCK epithelial cells in a cell density-dependent manner [25,26].

### 2.2. Cell-to-Cell Contact Interaction

The idea that signaling pathways mediate CIP was hypothesized because of the observation that YAP is localized in the nucleus when cell cultures reach confluence [27]. Unlike single-celled organisms, cell growth and division in metazoans with multicellular systems are inhibited by contact inhibition and the interaction between growth factors. The culture of human cells limits cell proliferation and division when cells contact each other, regardless of active cellular metabolism or the presence of growth factors. Contact inhibition is overcome in rapidly growing tissues during embryonic development, morning regeneration, wound healing, and uncontrolled proliferation and growth due to the loss of contact inhibition, which are also characteristics of solid tumors. E-cadherin is an example of such a contact inhibitory factor [7]. Cadherins inhibit cell contact-dependent proliferation by altering receptor tyrosine kinase signaling through interactions with the neurofibromatosis 2 (NF2) tumor suppressor Merlin. Additionally, the expression of E-cadherin in cells deficient in cadherin induces transportation of YAP into the cytoplasm from the nucleus. Thus, E-cadherin inhibits the nuclear localization of YAP [8,28]. Merlin also regulates the Hippo signaling pathway through its interaction with Kibra [29,30,31]. Merlin and Kibra act upstream of STK3/4, form complexes in cultured cells, and coexist in vivo [29,32]. These observations suggested that the Hippo signaling pathway can be regulated both in cultured cells and in vivo systems.

### 2.3. Hippo Signaling in Regulation of Cell Density and Cancer

Normal cell growth is restricted through cell–cell contact inhibition, regulating proliferation, whereas cancer cells continue to grow, by passing cell–cell contact inhibition [33]. However, cancer cells do sense the contact with surrounding cells and respond through signaling pathways. Deregulation of YAP signaling, which promotes cell growth and cell survival, was found to be upregulated in several cancers, and the activation or inactivation of YAP was correlated with poor prognosis of cancers [34,35,36]. At low cell densities, YAP showed decreased phosphorylation, increased nuclear localization, and upregulation of YAP target genes in cells. Reduced metastasis is consistent with YAP localization in the nucleus, and is representative of the invasive phenotype of cancer cells [37]. The loss of E-cadherin, Ca^2+^-dependent mediators of cell adhesion, and the formation of the cell junction promote tumor progression and metastasis [38,39]. The Hippo pathway was shown to be associated with E-cadherin-dependent contact proliferation inhibition. Depletion of LATS1/2 leads to E-cadherin ligation, thus suppressing E-cadherin-induced reduction in cell proliferation [7]. Deletion of LATS1 turns non-invasive cancer cells grown at high densities (trans-endothelial invasion), similar to those grown at low densities [37].

### 2.4. STK4 in Apoptosis

Apoptosis is a pathway centered on caspases [40]. STK3/4, which is crucial to the Hippo pathway, is cleaved and activated by caspases in response to apoptotic signals. The overexpression of STK4 induces caspase activity and activates the p38 MAPK and stress-activated protein kinase (SAPK) pathways, which are associated with apoptosis [41]. Therefore, STK4 acts as a positive regulator of the apoptotic response. Additionally, the basophilic kinase STK4 with pro-apoptotic activity is phosphorylated by the eosinophilic casein kinase 2 (CK2, also called CSNK2A1), which is anti-apoptotic [42,43]. Inhibition of CK2 activity induces apoptosis in cancer cells [44]. Cleavage of substrates by caspases is regulated by phosphorylation near the shear site. Serine 320 of STK4 is located close to the site cleaved by caspase, and CK2 phosphorylates serine 320 [45]. CK2 binds to STK4, including full-length STK4 located in the cytoplasm, and the N-terminal fragment of STK4 is located in the nucleus [43]. The interaction between the cytoplasm and nucleus is expected to open new avenues for downstream target phosphorylation.

STK3/4 regulates cell proliferation and apoptosis by targeting the cell cycle regulator cyclin E and apoptosis inhibitor DIAP1 [3,14,46,47,48]. In Drosophila, Hippo (Hpo, a homolog of mammalian STK3/4) induces apoptosis during development while promoting the proper termination of cell proliferation [14]. Hippo mutation leads to unregulated proliferation, resulting in enlarged tissue size as well as delayed cell cycle termination and resistance to pro-apoptotic signals, resulting in increased expression levels of cyclin E and DIAP1 [3].

STK4 is poorly expressed in human thyroid cancer cells; STK4 silencing promotes the proliferation of these cancer cells, inhibits apoptosis and autophagy, and thereby enhances tumor growth [49]. In addition, the overexpression of STK4 in MDA-T32 cells, a thyroid cancer cell line, induces cancer cell death by increasing mitochondrial stress, similar to that observed in YAP knockdown. The synergy between STK4 upregulation and YAP inhibition damages mitochondria, owing to the excess ROS production, and further increases apoptosis by activating the c-Jun N-terminal kinase (JNK) pathway [50]. Natural killer/T-cell lymphomas, a type of malignancy, exhibit downregulated expression of STK4, markedly upregulated expression of YAP, and suppressed phosphorylation of YAP. STK4 overexpression or YAP knockdown in natural killer/T-cell lymphomas is consistent with increased BCL2 and BAX expression levels, decreased cyclin D1 and CTGF expression levels, inhibition of cell proliferation and cell cycle progression, and promotion of apoptosis [51]. This suggests that the activation of the Hippo pathway can inhibit tumorigenesis.

### 2.5. Rules of Hippo Signaling in an In Vitro System

As described previously, the expression of nuclear-localized YAP is prevalent in cells cultured at low density and the activation of the Hippo pathway and cytoplasmic localization of YAP are prevalent in cells cultured at high density [18]. However, regardless of the density of the cultured cells, when serum was removed from the medium, YAP remained in the cytoplasm, and YAP(S127) phosphorylation dramatically increased. The absence of serum in the medium induced YAP(S127) phosphorylation and cytoplasmic maintenance independent of cell-cell contact. In contrast, the addition of serum to the medium induced YAP dephosphorylation and nuclear localization, ignoring cell-cell contact-mediated YAP inhibition. Serum substances that cause these responses include sphingosine-1-phosphate (S1P) and lysophosphatidic acid (LPA) [52]. S1P is a component of follicular fluid that regulates ovarian angiogenesis [53,54].

LPA is a simple phospholipid with various physiological and pathological functions, such as cell proliferation and differentiation [55,56], intercellular interactions [57], and tumorigenesis [58], and can be produced by endometrial and ovarian cells [59,60,61,62,63]. S1P activates YAP/TAZ and promotes cell proliferation by inhibiting LATS1/2 through the S1P receptor (S1PR) [52,64]. In zebrafish progenitor cells, YAP and CTGF are involved in the regulation of endoderm formation as downstream targets of S1P [65]. In mammals, nuclear localization of YAP is induced by S1P, increasing the CTGF transcript [66,67,68,69,70]. In addition to S1PR, S1P and LPA activate YAP/TAZ by inhibiting LATS1/2 via G protein-coupled receptors (GPCRs). LPA has six receptors capable of binding to GPCRs and can induce or inhibit YAP phosphorylation depending on which GPCR receptors are expressed and bind to the cell type [10].

## 3. Hippo Signaling Factors in the Uterine Endometrium

### 3.1. STK3/4 in the Uterine Endometrium

The Hippo signaling pathway cascade is triggered in the uterine epithelium to regulate downstream targets that affect dynamics during the estrous cycle. Interestingly, we found that the expression levels of STK3/4 change dynamically in the endometrium during the estrous cycle, demonstrating that STK3/4 is regulated by estrogen and estrogen receptors. STK3/4 is rapidly expressed via estrogen receptor-dependent signaling in both the endometrial glandular epithelium and luminal epithelium. STK3/4 mRNA and protein levels in the mouse endometrium were highest at 6 h of estrogen treatment and decreased thereafter. The regulation of STK3/4 and its downstream targets is mediated by p38-MAPK, an estrogen receptor-dependent intracellular signaling pathway. Our previous study showed that the phosphorylation of LATS1/2 was faster than that of STK3/4, which may occur through an unknown signaling pathway for the phosphorylation of LATS1/2 in uterine dynamics during the estrous cycle [71].

### 3.2. LATS1/2 in the Uterine Endometrium

Since its discovery in Drosophila, LATS1/2 has been considered a tumor suppressor. LATS1/2 plays an important role in controlling tumor development and the cell cycle through several mechanisms, including p53 [30,72,73,74]. 

However, there are few studies on the role of LATS1/2 in the uterus, among other Hippo signaling factors. In 2010, Strakova et al. reported an abstract describing that significant upregulation of LATS1/2 occurs during the secretory phase of the menstrual cycle in baboon (unpublished data). Additionally a recent study showed that *Scrib* knockout mice showed weak LATS1 expression in the uterus [75]. In addition, various mutations in LATS1/2 were found in human cancers, including uterine endometrial carcinoma [76]. 

The direct association between LATS1/2 and uterine endometrial cancer requires further research, but recent data implied that LATS1/2, like other Hippo factors, plays a role in regulating the uterine environment.

### 3.3. YAP in the Uterine Endometrium

YAP has been studied in endometrial stromal cells and deciduous membranes [77]. The desiccation of endometrial stromal cells is important for successful transplantation and maintenance of pregnancy, and it involves extensive cell proliferation and differentiation. YAP is upregulated in human decidual tissue, and YAP expression level in decidual cells is higher than that in the endometrial stromal cells of non-pregnant women [77]. The expression level of YAP mRNA is higher in the endometrial stromal cells in women with endometriosis compared with that in women without endometriosis [78,79,80]. mTOR is also overexpressed in eutopic endometrial stromal cells, indicating decreased autophagy. YAP expression influences cellular autophagy by mediating the mTOR-autophagy pathway in eutopic endometrial stromal cells. Treatment of isotopic endometrial stromal cells with the mTOR inhibitor rapamycin improved autophagy and significantly reduced YAP expression. This suggested a negative regulatory relationship between YAP and autophagy in eutopic endometrial stromal cells [78]. Thus, YAP may be considered as a biomarker for the treatment of endometriosis.

### 3.4. P190A in the Uterine Endometrium

Recent studies have reported a correlation between P190A and YAP in endometrial cancer [81]. Endometrial cancer is the sixth most common cancer in women worldwide and causes approximately 30% of deaths annually [82]. ARHGAP35, the gene encoding P190A, is mutated in approximately 15% of endometrial cancers. Inactivation of P190A due to mutations causes abnormal activation of YAP, a target gene of the Hippo signaling pathway, leading to the malignant transformation of cancer. When P190A is depleted in endometrial cancer cells, epithelial-mesenchymal metastasis (EMT) occurs, conferring metastatic function by enhancing resistance to cancer cell migration, invasion, and apoptosis. In P190A-depleted cells, the levels of CYR61, CTGF, THBS1, and ANKRD1 mRNA, the transcriptional target genes for EMT-related genes, and YAP were significantly increased. Moreover, a decrease in LATS1 and YAP phosphorylation was observed, suggesting that P190A negatively regulates YAP activity [81,82]. Therefore, depletion of P190A in endometrial cancer cells indicates that EMT is partially induced through the transcription of YAP in the Hippo pathway (Figure 3).

## 4. Downstream Targets of Hippo Signaling in the Endometrium

When LATS1/2 is inactive, YAP translocates to the nucleus without being phosphorylated. As YAP does not contain a DNA-binding domain, it binds to a transcription factor of the TEAD family, such as TEAD1-4, and induces the expression of target genes such as CTGF and CYR61 [83] that promote cell growth, proliferation, cell migration, and survival [35,84,85,86,87]. Other YAP target genes included AMOT, AMOTL1, AMOTL2 [88,89], CTNNA1 [90], CTNNB1 [91,92,93], ADAMTS1 [94,95], ANKRD1 [96,97], AXL [98], MCL1 [99], and THBS1 [100].

CYR61 is a marker of estrogen activity in normal and abnormal endometrial cells. In endometrial cancer, overexpression of CYR61 increases apoptosis and is suggested to be involved in growth arrest or growth stimulation depending on cell conditions [101,102]. AMOT, AMOTL1, and AMOTL2 are angiogenesis-related proteins regulated by estrogen in the luminal epithelium and progesterone in the endometrial matrix [88,89]. CTNNB1 is abnormally expressed in uterine fibroids and has been suggested to play a role in menstruation and implantation in the uterus. It has also been associated with endometrial hyperplasia and endometrial cancer; in low-grade endometrial cancer, CTNNB1 mutations are associated with recurrence-free survival [91,92]. ADAMTS1 is involved in various biological functions, including ovulation and embryo receptivity, through cell-matrix remodeling, developmental processes, or both, and is involved in bovine endometrial remodeling, which is required in conjunction with ovarian steroid hormones, for implantation and placental development [94,95].

### 4.1. Connective Tissue Growth Factor (CTGF)

CTGF, also known as cellular communication network factor 2 (CCN2),is a multifunctional growth factor that exhibits biological and immunological activities similar to that of platelet-derived growth factor (PDGF) and is strongly expressed in wound healing, angiogenesis, and connective tissue formation sites [103,104]. Therefore, CTGF may play a role in endometrial repair. CTGF expression was found to be consistent with endometrial repair and regeneration. CTGF is localized in glandular epithelial cells and macrophages in the stromal cell compartment, and CTGF mRNA and protein expression levels increase considerably in endometrial cells treated with progesterone and with hypoxia [105]. A marked increase in adhesion molecules, including CTGF, in the functional layer of the endometrium, suggests that CTGF influences the healing of the endometrial surface after menstruation. Restoring the epithelial surface without blood vessel repair in the endometrial epithelium will only result in the collection of a large amount of epithelial blood. Although the role of macrophages in endometrial epithelial recovery is not fully understood, they have been suggested to play an important role in endometrial growth and blood vessel formation [106]. Endometrial macrophages respond to estrogen and progesterone and are thought to contribute to the increased expression level of CTGF in stromal cells during menstruation when estrogen and progesterone levels are low. In ovariectomized mice, CTGF levels were increased by estrogen and decreased by progesterone. CTGF mRNA levels and protein localization in the uterine tissues of mice and pigs during the estrous cycle and early pregnancy have been studied along with spatiotemporal changes [107,108,109]. In the uterine fluids of pregnant pigs, CTGF concentration reached its highest level at the time of the generation of betrayal kidney and estrogen [103]. These data suggests that CTGF plays an important role in the estrous cycle, menstrual cycle, and pregnancy.

CTGF is located within the endometrial and decidual membranes, in a cell- and stage-specific manner, and in human endometrial tissue [101,105,110,111]. CTGF potentially regulates various normal cellular processes in uterine tissues. The localization of CTGF in mice differs depending on the stage of gestation. About 2.5 d after gestation, CTGF colocalized with TGFβ in uterine epithelial cells, but after about 4.5 d, there was a decrease in epithelial cells and an increase in stromal and endothelial cells. These changes occurred at different stages of extracellular matrix remodeling and angiogenesis in the endometrium. CTGF and TGFβ expression levels increased in the uterine epithelial cells due to estrogen induction and in the stroma. These effects were antagonized by progesterone; however, progesterone also appeared to temporarily induce CTGF production [112]. Therefore, CTGF in the mouse uterus is regulated by estrogen and progesterone via a TGFβ-dependent or independent mechanism.

### 4.2. Cysteine-Rich Angiogenesis Inducer 61 (CYR61)

CYR61, also known as CCN1, is an extracellular matrix (ECM)-related signaling protein from the CCN protein family. CYR61 is a multifunctional protein with a variety of functions, including cell adhesion and migration, cell proliferation and differentiation, and cell activation and apoptosis. It has conflicting roles as it both promotes tumor growth and inhibits tumor formation under various conditions. Additionally, it contributes to inflammation control, wound healing, and tissue repair [113,114,115,116]. CYR61 expression is downregulated in uterine leiomyosarcoma cells compared to normal uterine smooth muscle cells. Investigation of the factors affecting CYR61 in myometrial smooth muscle cells revealed that basal fibroblast growth factor (bFGF) and estrogen elevate CYR61 levels to heal uterine smooth muscle cells; progesterone has no effects on CYR61 expression [117]. In addition, the expression level of CYR61 is increased by EGF (epidermal growth factor) in endometrial epithelial cells. CYR61 was expected to be upregulated by the MAPK/ERK pathway triggered by the epidermal growth factor receptor (EGFR); however, MAPK/ERK showed no effect on CYR61 expression. JAK2/STAT3 pathway as an EGF sub-pathway, affects CYR61 expression. Blockage of the JAK2/STAT3 pathway leads to decreased CYR61 expression level [118].

CYR61 expression in the human endometrium promotes the proliferative phase of the menstrual cycle, in endometrial epithelial cells and the synovial endothelium [119,120]. It increases equally with VEGFA, an angiogenic factor upregulated during the menstrual cycle, suggesting a similar role in human endometrial vascular repair, growth, and maturation after menstruation [101]. CYR61 occurs downstream of the ERK pathway and is involved in angiogenesis induction [121]. CYR61 expression level was reduced in the endometrial adenocarcinoma cell line, Ishikawa compared to in normal endometrial cells, which is known to promote apoptosis of endometrial cancer cells and increase the expression levels of apoptosis-related genes. Treatment of Ishikawa cells with siCYR61 increased the cell count [122,123,124]. CYR61 was shown to have a progressive decline in expression level with the progression of endometrial adenocarcinoma from low-grade to high-grade. Therefore, CYR61 may be important for the development of endometrial cancer and survival prognosis [116,125].

### 4.3. Thrombospondin-1 (THBS1)

THBS1, also known as TSP or TSP-1, affects the structure of the extracellular matrix. It is an important factor in angiogenesis and tissue remodeling. Mutations in oncogenes and tumor suppressor genes in tumor cells are associated with decreased THBS1 levels [126]. As tumors grow, they require angiogenesis [127], which is induced by angiogenesis regulators such as VEGF. THBS1 is an endogenous angiogenesis inhibitor that inhibits tumor growth by inhibiting tumor angiogenesis [128]. THBS1 inhibits tumor cell growth by activating TGFβ in TGFβ-responsive tumor cells [126]. Tumor growth and angiogenesis were considerably increased in THBS1 cKO mice and in dilated capillaries. Conversely, in THBS1 overexpressing mice, tumor growth is delayed or underdeveloped and capillaries in the tumor are constricted [129].

The expression level of THBS1 in the human endometrium is significantly higher during the secretory phase than during the proliferative phase of the menstrual cycle. As in the mouse model, the number of capillaries is higher in endometrial cancer with high THBS1 expression levels, suggesting that elevated THBS1 expression is associated with an angiogenic phenotype in endometrial cancer [130]. As previously described, the potential effects of THBS1 and endometrial estrogen on inducing vascular destruction were evaluated. Estrogen treatment on the endometrium of baboons considerably decreased THBS1 expression levels in the endometrial glandular epithelial cells and stromal cells 6 h after estrogen injection. However, the expression level of angiopoietin, which promotes angiogenesis, increased in endometrial gland epithelial cells 3 h after estrogen treatment, and did not change even with decreased THBS1 expression level [131]. This finding suggested that estrogen induces rapid and cell-specific changes in the expression of angiogenesis-stimulating and angiogenesis-inhibiting factors in the endometrium.

### 4.4. Cyclin D1 (CCND1)

CCND1, also called BCL1, is a cell cycle regulator that can alter cell cycle progression and is essential for G1 phase progression [132]. CCND1 regulates proliferation by transducing extracellular signals [133]. It participates in the detachment (differentiation) of endometrial stromal cells via signal transduction [134]. CCND1 is a proto-oncogene with oncogenic characteristics in various carcinomas [135,136,137,138]. CCND1 is highly expressed in the proliferative and secretory phases of the menstrual cycle in the uterus [139], and its high expression level in endometrial cancer is observed in metastatic rather than non-metastatic carcinomas. This frequency of expression is significantly correlated with the stage of endometrial cancer; the higher the expression level, the poorer the prognosis of patients with endometrial cancer [140]. CCND1 expression is positively correlated with that of Ki67, which is a widely known marker of endometrial proliferation [141] and can be used as a marker for endometrial cancer cell proliferation [142]. The overexpression of CCND1 in endometrial cancer is associated with the loss of PTEN [143]. Loss of PTEN is commonly found in endometrial cancer and hyperplasia with cell dysplasia [144,145,146,147], which reinforces that CCND1 overexpression can be used as a marker for the early development of endometrial cancer.

## 5. Conclusions

The uterus undergoes dynamic changes during the reproductive cycle, including endometrial cell proliferation and decidualization induced by estrogen and progesterone. An imbalance in the levels of these hormones can lead to endometrial cancer or an abnormal uterine environment. It is becoming increasingly clear that Hippo signaling pathway, which regulates angiogenesis, cell proliferation, differentiation, and apoptosis, is involved in the changes caused by the uterine cycle. Our review elucidates the dynamic changes in the endometrium influenced by the Hippo signaling pathway.

## Figures and Tables

**Figure 1 ijms-23-03852-f001:**
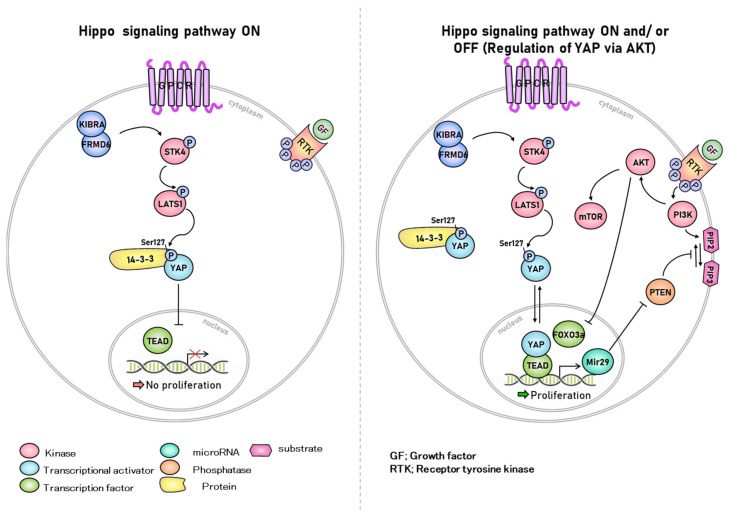
**Hippo signaling pathway.** YAP, a downstream target of the Hippo signaling pathway, activates the mammalian target of rapamycin (mTOR), a major regulator of cell growth. This suggests that the Hippo pathway, PTEN, and AKT are related. The stimulation of AKT and inhibition of the Hippo signaling pathway affect ovarian follicle growth. AKT activates dormant primordial follicles and promotes follicle growth by interfering with the Hippo signaling pathway by ovarian fragments. YAP regulates carcinogenesis in endometrial adenocarcinoma, and YAP expression correlated with the type of endometrial adenocarcinoma. In addition, the Hippo signaling pathway is a key downstream signaling branch of the G protein-coupled estrogen receptor (GPER) and plays a crucial role in breast tumorigenesis.

**Figure 2 ijms-23-03852-f002:**
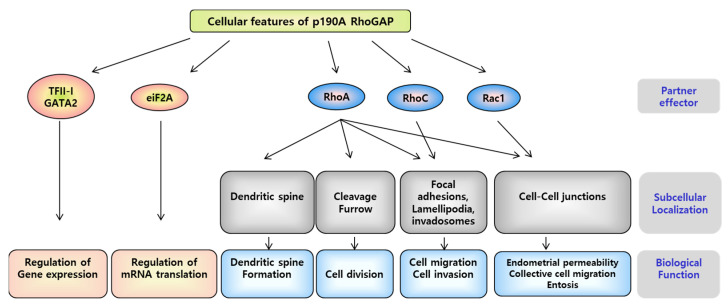
**Role of P190A RhoGAP in various cellular processes.** The interaction between P190A and p120-catenin at the cell-cell junction inhibits RhoA activity. CRAD (catenin-RhoGAP-association domain), a C-terminus domain of p120-catenin, interacts with P190A. CRAD depletion in p120-catenin of endothelial cells affects the transmembrane transduction of P190A and regulates endothelial permeability by increasing RhoA and decreasing the Rac1 signal. The crossover between cell-cell contact and cell-matrix adhesion is produced by the P190/P120-catenin complex to spatially regulate RhoA activity. The transient decrease in P190A levels during late mitosis in the cell cycle finely regulates RhoA activity required for cell division; P190A is also required to regulate spindle formation [23].

**Figure 3 ijms-23-03852-f003:**
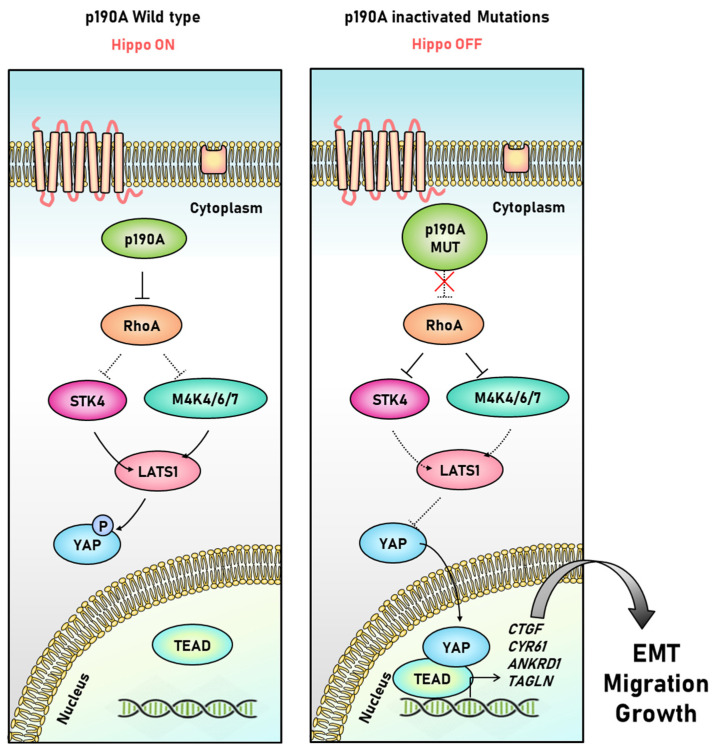
**I****nteraction between P190A and Hippo signaling pathway in endometrial cancer cells.** Under normal conditions, P190A induces YAP phosphorylation by crossing the Hippo signaling pathway through RhoA inhibition. When P190A is mutated, RhoA is abnormally activated, interfering with the activity of STK4 and MAP4K4/6/7, preventing the phosphorylation of LATS1/2, and triggering the transcriptional activation of YAP, resulting in the transcription of EMT-related target genes [81].

## Data Availability

Not applicable.

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
