# Peer review of "Hippo Signaling in the Endometrium"

_ijms, 2022, doi:10.3390/ijms23073852_

Round 1

Reviewer 1 Report

This is a well written review article dealing with the relevance of the Hippo signaling pathway the female reproductive system focusing on different common pathologies

Besides some minor spell changes and if relevant the check for most recent literature I do not have additional comments to make.

Author Response

I would like to thank you for your precious comments.

Please find an attached file.

Reviewer 2 Report

the text contains very interesting topics; the authors used an extensive bibliography. In my opinion, to make the text more readable to readers, the authors could insert tables within the more substantial paragraphs. It is also advisable to carry out an English language check

Author Response

(The authors gave the same response as above.)
